# The Impact of Short-Term Formalin Fixation on Weight and Ventricular Dimensions in the Hearts of Cats and Small-to-Medium-Sized Dogs [note 1]

**DOI:** 10.3390/vetsci12010074

**Published:** 2025-01-20

**Authors:** Izabela Janus-Ziółkowska, Joanna Bubak

**Affiliations:** Department of Pathology, Wrocław University of Environmental and Life Sciences, 50-375 Wrocław, Poland; joanna.bubak@upwr.edu.pl

**Keywords:** canine, feline, cardiac, cardiac measurements, referral opinion, forensic examination

## Abstract

Post-mortem studies help to provide a diagnosis in clinical and forensic cases. They can be performed in-house, although in challenging cases, they should be referred to a specialised laboratory. Cardiac disease, including hypertrophy, is one of the circumstances that requires a referral opinion. The low number of units specialising in cardiac pathologies requires the shipment of formalin-fixed hearts. Studies on different tissues (brain, placenta, prostate gland, and various types of neoplasms) show that short- and long-term formalin fixation can affect both tissue weight and dimensions. No such study has been conducted on the short-term formalin fixation of feline and canine hearts. We aimed to describe the impact of short-term fixation on cardiac morphometry. Based on our study, we found that a fixation up to 72 h does not influence the heart weight and dimensions in cats and small-to-medium-sized dogs. Nonetheless, the proper flushing and drying of the organ is essential for reliable results. Our study supports the possibility of the shipment of hearts to referral centres to obtain the opinion of a specialist, especially in complex cases.

## 1. Introduction

A detailed cardiac morphometric analysis is a part of post-mortem and forensic examination, especially in cases of sudden death or suspicion of cardiac hypertrophy of various origins [1,2]. Due to the organ’s complexity, in humans, it is recommended that the examination be performed by experienced specialists [1,2]. In veterinary medicine, no similar recommendations exist even though there are species, breed, and size differences that require a post-mortem examination by a specialist. The availability of doctors specializing in cardiac post-mortem examination is low in both human and veterinary medicine; therefore, it is also recommended to send the heart obtained post-mortem to a specialist for a referral opinion [1].

A standard shipment procedure for the specimen involves the fixation of the obtained heart in 6–8% formalin solution. It has been observed that 1-year fixation can affect both the heart weight and heart dimensions in porcine hearts [3]. Simultaneously, long-term fixation also affects other tissue features (e.g., light absorbance or antigen availability used in either polarised light microscopy or immunohistochemistry [4,5,6]). Therefore, in terms of post-mortem diagnostics or forensic examination, the fixation time should be kept as short as necessary for tissue fixation and shipment, and—if possible—not exceed 72 h to guarantee a proper final diagnosis [5].

While the heart’s weight and dimensions serve as confirmation criteria for the ante-mortem and post-mortem diagnosis of various cardiac diseases in dogs and cats, e.g., hypertrophic cardiomyopathy or dilated cardiomyopathy [7,8], the impact of short-term formalin fixation on feline and canine heart measurements has not been previously evaluated.

The reaction of a tissue to formalin fixation depends on the type of tissue. So far, studies have been conducted on brains, placentas, and various types of tumours, including ones affecting the prostate gland, lungs, kidneys, or the head and neck. Various research teams have observed either thickening and weight gain or shrinkage and weight loss in the examined tissue [9,10,11,12,13,14,15,16,17,18,19,20]. The tissue reaction to formalin fixation also differs depending on the part of the organ [17,20]. At the present time, the reason for various tissue responses is not known. Therefore, the results obtained in one organ cannot be extrapolated to other organs.

Simultaneously, the tissue reaction to fixation changes with time [18,20]; therefore, one cannot generalise the results obtained in a long-term fixation of the heart [3] on the tissues examined within a few days of fixation.

The aim of this study was to evaluate the effect of short-term (72 h) formalin fixation on cardiac weight and ventricular dimensions in cats and small-to-medium-sized dogs to determine whether there is a statistically significant impact from processing. We hypothesise that formalin fixation has no significant impact on cardiac tissue weight and dimensions in cats and small-to-medium-sized dogs.

## 2. Materials and Methods

The study was performed on hearts obtained post-mortem from cats and dogs. All tissues were collected during teaching necropsies, and no animal was sacrificed deliberately for the study. The animals’ owners gave their consent for the use of the bodies in the teaching process.

During a teaching necropsy, performed by students within 24 h of the animal’s death or euthanasia, the body is weighed at the beginning of the examination using a standard scale with a precision of 0.1 kg. The skin is dissected in the midline from the chin towards the pelvis. Next, the skin is dissected from the subcutaneous tissue over the thoracic and abdominal cavities. Subsequently, the abdominal cavity is opened along the *linea alba*, and the thoracic cavity is opened by cutting the ribs on the costochondral junctions. After removing the sternum, the thoracic cavity is exposed. The organs of the mouth, neck, and thoracic cavity are removed from the body after a V-shaped incision is made medially to the mandible and the dissection of organs from the spine. Finally, the organs are separated from the diaphragm by cutting the caudal vena cava, oesophagus, aorta, vagal nerve, and phrenopericardial ligament.

After the dissection performed in a routine above-mentioned technique, the pericardial sac was totally removed, heart was cut off from the lungs directly above the origin of the great arteries and the weight of the heart was determined to the nearest gram (Figure 1 and Figure 2B).

In the next step, the hearts were cut transversally (perpendicular to the cardiac long axis) through the ventricles below the atrioventricular valves, carefully flushed with running water to remove all remaining blood, padded dry with a surgical towel to remove any residual fluid [21], and weighed again (time point 0; Figure 2C,D). Next, the dimensions of the ventricles (left ventricular posterior wall thickness: LVPW; left ventricular internal diameter: LVID; interventricular septum thickness: IVS; right ventricular internal diameter: RVID; right ventricular wall thickness: RVW) were measured using a manual Vernier calliper to the nearest 0.1 mm (time point 0; Figure 1; Figure 2D), as described previously [22]. The measuring lines were perpendicular to the cardiac walls and localised in the middle between the papillary muscles, as depicted in Figure 2D. The measurement points were marked with a tissue marker (Figure 1) to ensure consistency in choosing the points of the measurements. All the measurements were taken without additional pressure on the tissue to avoid its compression. After the measurements, the hearts were immersed in a 7% buffered formalin solution. After 24, 48, 72, and over 72 h (from 96 to 144 h), each heart was collected, dried, weighed, and measured as in time point 0 (Figure 2E,F).

Each measurement was obtained by both authors independently (IJZ and JB). In case of a difference exceeding 5% for the same measurement, the measurements were repeated. Each measurement was repeated two times by each researcher, and a mean value from all measurements was calculated.

Hearts that underwent processing errors due to the teaching character of tissue collection were excluded from further analysis (either weight comparison in cases of improper flushing or dimension comparison in cases of improper dissection angle).

A heart-weight-to-body-weight ratio was subsequently calculated for both intact hearts and organs after flushing and drying.

All obtained results were recorded and subjected to statistical analysis. The analysis was performed using the Statistica 13.3 software (Tibco, Cracow, Poland) and appropriate tests. Data normality was confirmed using Shapiro–Wilk analysis. Differences between groups were obtained using either Student’s *t*-test (for heart weight and heart-weight-to-body-weight ratio before and after flushing of blood and drying) or one-way repeated measures ANOVA analysis with the Tukey post hoc test (for heart weight and dimension changes over time). The significance was set at *p* ≤ 0.05.

## 3. Results

Organs enrolled in the study included 134 canine and feline hearts. Among them, 95 hearts were obtained from cats (body weight: 1.5 kg to 10 kg), and 39 hearts were obtained from dogs (body weight: 2.2 kg to 17 kg). Due to the teaching character of the necropsy and the involvement of the students in tissue handling, the blood was partially flushed from the heart prior to the first weighing in 44 hearts; therefore, 90 hearts (68 feline and 22 canine) served to compare heart weight prior to and after flushing. Due to the same reason, in 60 hearts, the dissection was made with an incorrect angle (not perpendicular to the cardiac long axis); therefore, 74 hearts (52 feline and 22 canine) served to compare the ventricular dimension changes over fixation time.

The mean and standard deviation for heart weight prior to and after flushing, heart-weight-to-body-weight ratio prior to and after flushing, and ventricular dimensions at the examined time points for the cats and dogs are presented in Table 1 and Table 2.

The hearts were significantly heavier prior to blood removal than after flushing and drying in the whole group (Student’s *t*-test, *p* < 0.0001), in cats (Student’s *t*-test, *p* < 0.0001; Figure 3A, Table 1), and in dogs (Student’s *t*-test, *p* < 0.0001; Figure 3A, Table 2), but the weight did not change within the fixation time (ANOVA analysis; *p* > 0.05 for all comparisons; Figure 4, Table 1 and Table 2). Similarly, the heart-weight-to-body-weight ratio differed significantly prior to and after blood flushing in the whole group (Student’s *t*-test, *p* < 0.0001), in cats (Student’s *t*-test, *p* < 0.0001; Figure 3B, Table 1), and in dogs (Student’s *t*-test, *p* < 0.0001; Figure 3B, Table 2)

Similar to heart weight, no ventricular dimensions changed significantly within the fixation time (ANOVA analysis: *p* > 0.05 for all comparisons; Figure 5, Table 1 and Table 2), although a mild increase in ventricular wall thickness and a decrease in ventricular internal diameters were noted, especially within the first 24 h of fixation.

## 4. Discussion

Our results show that although the heart weight in cats and small-to-medium-sized dogs was affected by blood flushing and tissue drying, days-long formalin fixation did not significantly influence either the heart weight or ventricular dimensions.

The difference in heart weight prior to and after blood flushing and tissue drying is not surprising; nonetheless, it has been shown that the method of heart emptying with or without drying significantly affects the human heart’s weight [21]. Despite the difference in heart sizes between humans and companion animals, in our study, the observed difference was not only statistically significant but also clinically significant. The border value for normal heart weights in cats is 20 g, and the normal value for the heart-weight-to-body-weight ratio is 3–4 g/kg [7]. In dogs, the normal value for the heart-weight-to-body-weight ratio has differed between researchers and can vary between 6.4 and 8.4 g/kg [23]. Therefore, an improper weighing protocol may lead to false-positive results in cardiac hypertrophy. Although in our study both normal and hypertrophied hearts were enrolled, the average weight of feline hearts after flushing was lower than the borderline (20 g), with an average heart-weight-to-body-weight ratio mildly elevated in both species. At the same time, the results prior to blood flushing were markedly above the border values for both cats and dogs, proving that the process of cleaning the heart may impact the results. Therefore, it should be clearly stated that in cases of suspicion of cardiac hypertrophy, heart weighing should be performed after a thorough flushing and drying of the organ. 

The effect of formalin fixation on organ weight and size varies depending on the organ type, organ part, and histological structure [11,12,13,17,18,24].

In a study on formalin-fixed mouse brains, Weisbecker [20] showed that the organ initially increases its weight (within the first 24 h by approx. 54%), only to slowly reverse the trend in the following days and reach the original organ weight after 213 days. A similar effect was noted in formalin-fixed placentas, but the weight gain was significantly lower than in the study on brains, and no reversal of this effect was observed within the next four days [18]. In our study, no changes in heart weight were noted within the observation period. Given the various histological structures of the brain, placenta, and cardiac muscle, the differences observed between the studies may be associated with the cellular and interstitial tissue density of the organs, but no such comparison has been performed.

The effect of formalin fixation on tissue size differs depending on the organ. In the majority of cases, formalin fixation causes shrinkage in the tissue [9,10,14,16,19,25]. Depending on the study, the changes in size can be statistically and/or clinically significant [14,19] or insignificant [9]. Interestingly, Weisbecker [20] found that the mouse brain initially increases in size and subsequently (after several days) shrinks. Similarly, in a study on tonsils, Vent et al. [24] noted that the organ’s length increases within the first 72 h. The changes, although statistically significant, were not clinically significant. In our study, the ventricular wall thickness increased within the first 24 h with a simultaneous reduction in ventricular internal diameters. However, the observed changes were insignificant both statistically and clinically. Moreover, the increase in wall thickness did not progress or regress in consecutive days.

The changes in formalin-fixed organ weight and tissue thickness also depend on the concentration of the formalin solution [26]. Low-concentration solutions (2% and 4%) cause a significant increase in tissue weight, while in higher concentrations (10%), this effect is not observed. In our study, we chose a 7% formalin solution, as this is the one commonly used in veterinary practice for the shipment of tissues. Nonetheless, it has to be remembered that over a long period of time (a year), the cardiac parameters change regardless of the fixative concentration [3]. Moreover, the type of fluid used for tissue flushing, the type and concentration of the fixative, and the method of tissue processing may affect the tissue structure (especially the ultrastructure) and the possibility of performing histological, histochemical, immunohistochemical, and molecular examinations [27,28]. In the present research, we focused on the possible gross morphological changes that can occur during fixation with the protocol routinely used in private practices. Comparing various methods of tissue flushing and fixation, however interesting, was out of the scope of this paper.

One of the limitations of this study was the use of hearts obtained during teaching necropsies, which resulted in the need to exclude multiple samples from some comparisons due to students’ technical errors. Nonetheless, the use of teaching material reduced the need to sacrifice animals deliberately for the purpose of the study and, therefore, has not contributed to animal suffering.

The next limitation of our study is the use of a manual calliper. To provide reliable results, a Vernier scale calliper was used, and no additional pressure was applied to the tissues during measuring. It enabled us to show no clinical differences in cardiac measurements performed within the first days of fixation and fulfil the purpose of the study. Nonetheless, to study minimal changes in the gross measurements, especially combined with changes in histological structure and ultrastructure, a method using, e.g., the measurement of a specimen’s digital images, would be beneficial.

Another limitation is the use of only small-to-medium-sized dogs. In order to minimise animal suffering, this study was conducted on material used for teaching necropsies performed by the students. Small-and-medium-sized dogs are more common in our country than large- and giant-breed dogs. As a result, teaching necropsies for the latter are rare. Due to a small number of large- and giant-breed dogs examined by students during the study period, we decided not to include the hearts obtained from these dogs in the study. Although our results do not suggest that the difference in heart weight or ventricular dimensions could be more significant with greater heart weight, one should be cautious when extrapolating these assumptions on large- or giant-breed dogs.

The last limitation is a high diversity of heart weight in the whole study group and, specifically, in the canine group. This resulted from a wide range of animal sizes, and therefore—to omit that limitation—a comparison of heart-weight-to-body-weight ratios was performed.

## 5. Conclusions

In conclusion, short-term formalin immersion, enabling tissue fixation and possible shipment, does not affect heart weight and ventricular dimensions in cats and small-to-medium-size breed dogs. Conducting a detailed post-mortem cardiac examination is possible and reliable in cooperation with not-in-house reference centres. Nonetheless, blood flushing and heart drying are essential prior to weighing the heart and, when not performed, should be clearly indicated in the protocol.

## Figures and Tables

**Figure 1 vetsci-12-00074-f001:**
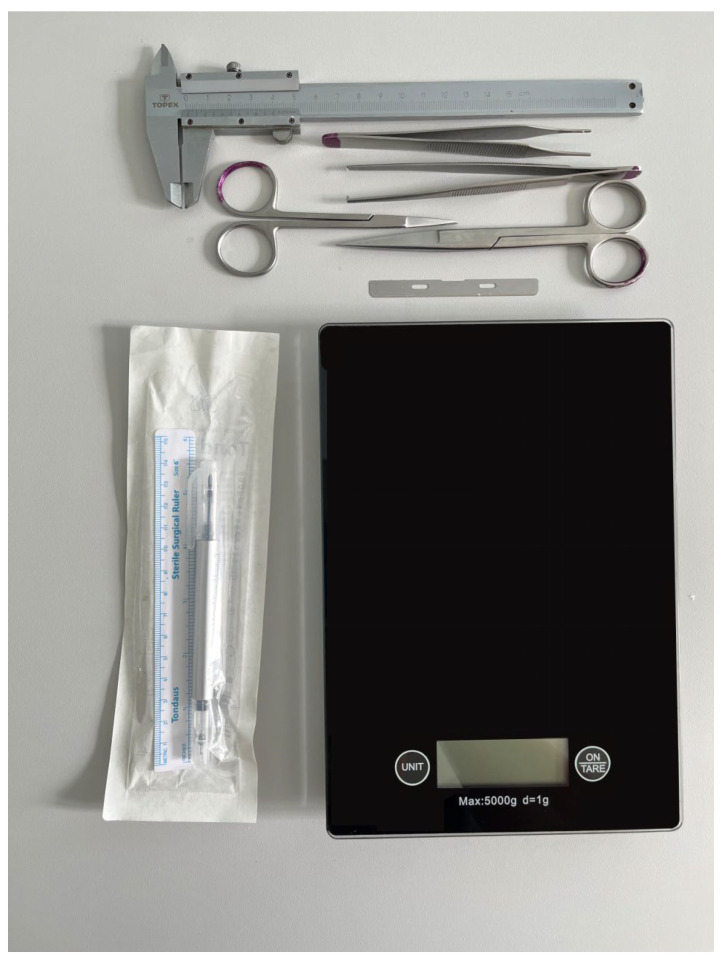
The laboratory equipment used in the study: laboratory scale with an accuracy of 1 g, manual Vernier calliper with an accuracy of 0.1 mm, scissors, forceps, microtome blade, and tissue marker.

**Figure 2 vetsci-12-00074-f002:**
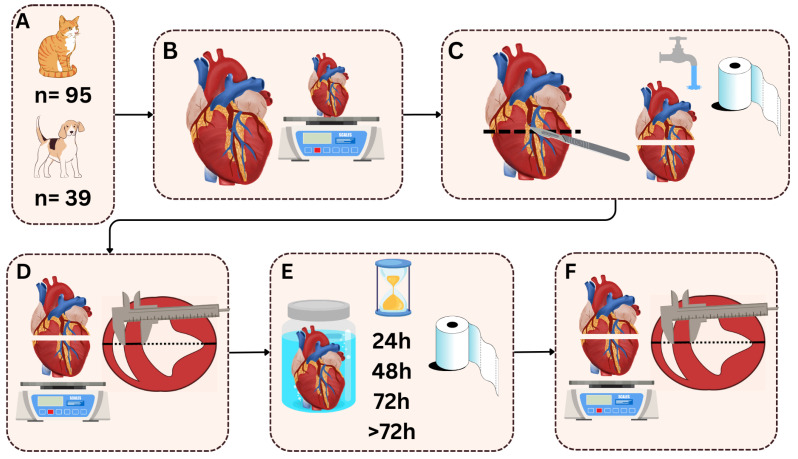
Methodology of the study. (**A**) Study population; (**B**) weighing of the intact heart; (**C**) heart dissection, flushing, and drying; (**D**) weighing and measurement of the dissected cleaned and dried heart; (**E**) immersion in a 7% buffered formalin solution and drying at study time points; (**F**) weighing and measurement of the heart after 24, 48, 72, and over 72 h.

**Figure 3 vetsci-12-00074-f003:**
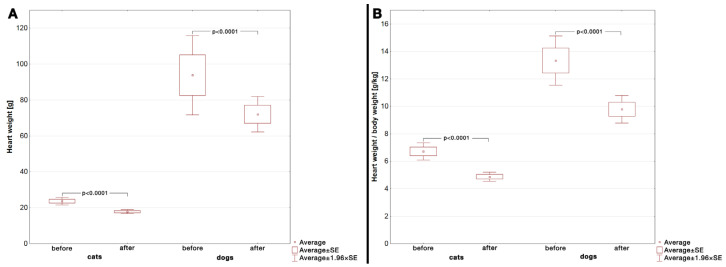
The heart weight (**A**) and heart-weight-to-body-weight ratio (**B**) before and after flushing of blood and drying the organ. The differences were significant in cats and in dogs (Student’s *t*-test: *p* < 0.0001 for all comparisons). SE—standard error.

**Figure 4 vetsci-12-00074-f004:**
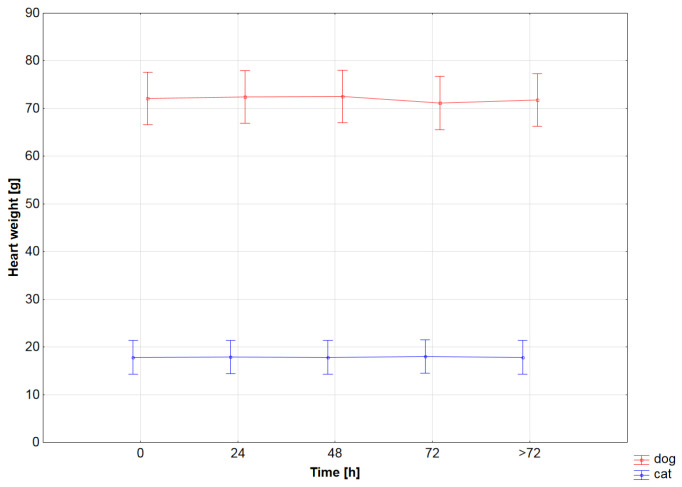
The heart weight in the examined time points showed no statistical difference in dogs (red) or in cats (blue). ANOVA analysis: *p* ≥ 0.05 for all comparisons. Results are presented as mean ± 95% confidence interval.

**Figure 5 vetsci-12-00074-f005:**
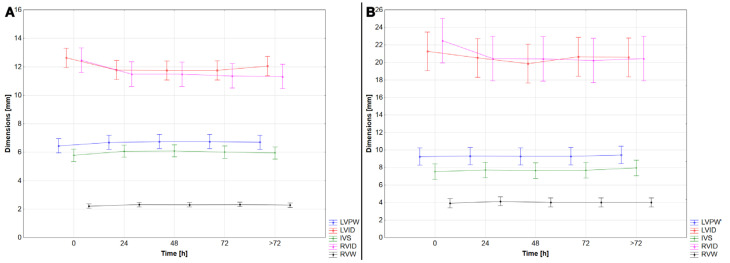
The cardiac dimensions in the examined time points showed no statistical difference in cats (**A**) or in dogs (**B**). ANOVA analysis: *p* ≥ 0.05 for all comparisons. Results are presented as mean ± 95% confidence interval. LVPW—left ventricular posterior wall thickness; LVID—left ventricular internal diameter; IVS—interventricular septum thickness; RVID—right ventricular internal diameter; RVW—right ventricular wall thickness.

**Table 1 vetsci-12-00074-t001:** Heart weight and measurements of cats at examined time points.

Timepoint [h]	0	24	48	72	>72
HPF [g] (*n* = 68)	23.8 ± 8.8 ^a^				
HPF/BW [g/kg] (*n* = 68)	6.7 ± 2.5 ^b^				
HAF [g] (*n* = 95)	17.8 ± 5.9 ^a^	17.9 ± 5.9	17.8 ± 5.9	18.0 ± 6.0	17.8 ± 5.8
HAF/BW [g/kg] (*n* = 95)	4.9 ± 1.6 ^b^				
LVPW [mm] (*n* = 52)	6.4 ± 1.9	6.7 ± 1.8	6.7 ± 1.8	6.7 ± 1.8	6.7 ± 1.8
LVID [mm] (*n* = 52)	12.6 ± 3.2	11.8 ± 2.3	11.7 ± 2.2	11.8 ± 2.2	12.1 ± 2.3
IVS [mm] (*n* = 52)	5.8 ± 1.6	6.1 ± 1.5	6.1 ± 1.6	6.0 ± 1.5	5.9 ± 1.6
RVID [mm] (*n* = 52)	12.5 ± 3.7	11.5 ± 3.0	11.5 ± 3.0	11.4 ± 3.0	11.3 ± 2.9
RVW [mm] (*n* = 52)	2.2 ± 0.6	2.3 ± 0.6	2.3 ± 0.6	2.3 ± 0.6	2.3 ± 0.6

The results are presented as mean ± standard deviation; HPF—heart weight prior to flushing; HPF/BW—heart-weight-prior-to-flushing-to-body weight ratio; HAF—heart weight after flushing; HAF/BW—heart-weight-after-flushing-to-body weight ratio; LVPW—left ventricular posterior wall thickness; LVID—left ventricular internal diameter; IVS—interventricular septum thickness; RVID—right ventricular internal diameter; RVW—right ventricular wall thickness. Statistically significant differences were noted only in the heart weight and heart-weight-to-body-weight ratio before and after organ flushing and drying. The differences are marked with the following superscripts: ^a^
*p* < 0.0001; ^b^
*p* < 0.0001 (Student’s *t*-test).

**Table 2 vetsci-12-00074-t002:** Heart weight and measurements of dogs at examined time points.

Timepoint [h]	0	24	48	72	>72
HPF [g] (*n* = 22)	93.8 ± 52.8 ^a^				
HPF/BW [g/kg] (*n* = 22)	13.3 ± 4.3 ^b^				
HAF [g] (*n* = 39)	72.1 ± 31.5 ^a^	72.5 ± 31.4	72.5 ± 31.2	71.2 ± 31.7	71.8 ± 31.1
HAF/BW [g/kg] (*n* = 39)	9.8 ± 3.2 ^b^				
LVPW [mm] (*n* = 22)	9.3 ± 2.6	9.3 ± 2.3	9.3 ± 2.3	9.3 ± 2.3	9.4 ± 2.1
LVID [mm] (*n* = 22)	21.2 ± 5.2	20.5 ± 5.2	19.9 ± 5.5	20.6 ± 5.2	20.6 ± 5.1
IVS [mm] (*n* = 22)	7.5 ± 1.8	7.7 ± 2.1	7.6 ± 2.2	7.7 ± 2.2	7.9 ± 2.1
RVID [mm] (*n* = 22)	22.5 ± 6.4	20.4 ± 5.6	20.4 ± 6.1	20.2 ± 5.8	20.4 ± 6.1
RVW [mm] (*n* = 22)	3.9 ± 1.2	4.1 ± 1.4	4.0 ± 1.2	4.0 ± 1.2	4.0 ± 1.1

The results are presented as mean ± standard deviation; HPF—heart weight prior to flushing; HPF/BW—heart-weight-prior-to-flushing-to-body weight ratio; HAF—heart weight after flushing; HAF/BW—heart-weight-after-flushing-to-body weight ratio; LVPW—left ventricular posterior wall thickness; LVID—left ventricular internal diameter; IVS—interventricular septum thickness; RVID—right ventricular internal diameter; RVW—right ventricular wall thickness. Statistically significant differences were noted only in the heart weight and heart-weight-to-body-weight ratio before and after organ flushing and drying. The differences are marked with the following superscripts: ^a^
*p* < 0.0001; ^b^
*p* < 0.0001 (Student’s *t*-test).

## Data Availability

The original contributions presented in this study are available after contact with the corresponding author.

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
