# Peer review of "The Impact of Short-Term Formalin Fixation on Weight and Ventricular Dimensions in the Hearts of Cats and Small-to-Medium-Sized Dogs"

_vetsci, 2025, doi:10.3390/vetsci12010074_

Round 1
Reviewer 1 Report
Comments and Suggestions for Authors
The revisions and comments are included in the manuscript PDF, and the report is included below.

Author Response
Dear Reviewer, Thank you very much for taking the time to review this manuscript. We believe that with the revisions made after all Reviewers' suggestions, the manuscript is improved. Please find the detailed responses below and the corresponding revisions/corrections highlighted red in the re-submitted files.The impact of short-term formalin-fixation on the weight and ventricular dimensions in the hearts of cats and small-to-medium-sized dogs
In this manuscript, the changes in weight and dimensions that occur in the hearts of dogs and cats when subjected to a short formalin fixation process (0-72 hours and up to six days) are studied. Hearts from cats and small to medium-sized dogs obtained from educational necropsies are used to determine if formalin-fixation can cause substantial changes that may interfere with the diagnosis of cardiac pathologies when they need to be sent to a reference center.
In general, no major flaws are observed in the manuscript. The work is well-organized, but it is unfortunate that starting from 134 hearts, the effective number of samples is considerably reduced. In cats, out of 95, only 52 were measured, and similarly in dogs, only 22 out of 39 were used to obtain data.
Thank you for that comment. We agree that it is a serious limitation of the study. Nonetheless, due to ethical reasons, we decided to use the teaching material with the consequences of students’ errors. As a result 90 hearts served for the comparison of the organ’s weight prior and after blood flushing and tissue drying and 74 hearts served for the comparison of ventricular dimensions. All hearts (134) served for the comparison of organ weight in study time points.
On one hand, in figure 2, considering that the hearts are from dogs and cats, i.e., carnivores, which share common cardiac morphology and the branches of the aorta (only two in the aortic arch: the brachiocephalic trunk and the left subclavian artery), a diagram with a human heart showing three branches emerging from the aortic arch is used instead. This needs to be corrected, and the diagrams should be slightly enlarged.
Thank you for that comment and pointing out our oversight. We have corrected the heart diagram to be representative for carnivores. We also enlarged the diagram’s components to show the details better.
On the other hand, regarding the data tables, as noted in the comments of the PDF file:
-Table 1, which presents data for both species combined, does not provide any new information compared to the species-specific tables 2 and 3. Therefore, I suggest eliminating Table 1.
We have removed Table 1 from the manuscript.
-The format of the tables can also be improved. The last items are abbreviated, and the first four items can also be abbreviated to unify the format. I suggest something similar to or improved from the examples I propose at the end of this report (from Table 2).
Thank you for the comment and suggestions. We abbreviated the names of first items.
-At the end of the tables it appears ap<0.0001; bp<0.0001, but there is no further explanation, so it would be advisable to better explain what they indicate.
We added appropriate explanation to the presented p-values
In addition, Figures 3-6 are very small and barely legible. Since they consist of three images: (a) corresponding to whole data; (b) data from cat hearts; and (c) data from dog hearts, and, as mentioned, the whole group data is dispensable, part (A) of these figures (Figure 3A; Figure 4A; Figure 5A; and Figure 6A) could be removed. Additionally, the size of parts (B) and (C) can be enlarged so that they can be read and viewed without difficulty.
Thank you for that comment - the image size is partially limited by the page size. We believe that in the final version the figures could be magnified. Nonetheless, we removed the part of the figures with the comparison of the whole group, as suggested. Also, we merged Figure 3B with 3C and 5B with 5C, and subsequently - merged Figure 3 with figure 5. We also removed Figure 4A and presented Figure 4B and 4C on one graph
Finally, regarding the bibliography, everything is correct as the cited bibliography appears in the references section, and all references (26) have been cited at least once.
Thank you for checking that point and confirming no mistakes.
Referred to the format of the tables:
Just as the names of the measurements taken in the heart are written in abbreviated form, the nomenclature of the first four parameters can also be abbreviated, making the table more uniform. These are some examples of Table 2:
As mentioned before, we changed the tables as suggested
L41 These terms are already in the title. You should select others to complement the bibliographic search, such as 'feline' instead of 'cat'; cardiac measurements
Thank you for that comment; we changed the keywords as suggested
L49 ''Veterinarians specialising in cardiac potsmortem ...both in HUMAN and veterinary medicine' sounds weird.
Thank you for pointing that out. We changed the sentence.
L57 Remove ;
Change made as suggested
L70 Remove (
Change made as suggested
Figure 2 In this diagram of the methodology of the study referring to the hearts of dogs and cats, a drawing of the heart of these carnivores should be used (with two branches emerging from the aortic arch) instead of the human heart diagram (with three branches emerging from the aortic arch).
In addition, although the diagrams are visible at this size, they could be enlarged slightly.
As mentioned before, thank you for that comment. We made changes as suggested.
Figure 2 The drawing of the diagram is too small to appreciate those details
The details of the drawing were enlarged
Tables 1-3 are cited before figures 3-5. Hence,Tables 1-3 should appear before figures 3-5
We moved the tables before the figures
Figure 3 Considering the group as a whole (A) does not provide any useful information. Hence, (A) could be removed and then, fuse B and C in the same chart -using different colour for cats and dogs-. This will allow you to increase the size of the chart and the font so that the legends could be read perfectly.
Figure 4 This figure should be placed after the representation of the heart-weight-to-body-weight. Information from the whole group (A) could be removed, and (B) and (C) merged in the same chart using different colour for each species.
Figure 5 This figure should be placed after Figure 3, as is the same representation although referenced to the body weight. I recommend doing the same as in Figure 3: remove (A) and fuse (B) and (C) in the same chart with different colours.
We made changes in the figures as described before
Table 1 I wonder if this table, including data from all the animals, has any importance, because the difference in heart size between cats and dogs is considerable .
I am unsure what additional information might be gained by considering all the samples together. I suggest removing Table 1 because Tables 2 and 3 give all the information needed.
We removed table 1 as suggested
Tables 2 & 3 The appearance of this table could be improved. Two examples are proposed.
We changed the tables as suggested
Tables 2 & 3: Replace “in” with “at”
Correction made as suggested
Tables 2 & 3 What do 'a',and 'b' stand for as they have the same significance? Are they from Student's t-test? Some clarifying information should be provided here or in the chapter of Material and Methods.
Thank you for that comment. We added additional explanations in the tables legends and also added information in the Materials and methods section.
Table 3: Remove the dot at the end of the line.
Correction made as suggested
Figure 6: You can omit diagram (A) and keep diagrams (B) and (C) as they are. Ensure that the legends have a font size large enough to be readable.
We made changes in the figures as described before
L211 mouse brain instead of mice brain
Correction made as suggested
L225 mouse brain
Correction made as suggested
L231 the wall thickness
Correction made as suggested
Reviewer 2 Report
Comments and Suggestions for Authors
Veterinary sciences 3351805
This paper describes changes in heart weight and cardiac wall and chamber dimensions in response to immersion in a 7% formalin solution and follows those changes for 72-96 hours. The objective of the investigation is to determine whether formalin preservation of cardiac tissues would materially impact findings for samples submitted for further investigation to referral centres. The paper is well-written, succinct and generally easy to follow. There are a number of items that are missing or inadequately described, however, while the focus of the paper and the authors' interpretations and conclusions seem to this reviewer to be more narrow than can be justified by the study's findings. Some information and the full implications of findings are missing.
It is not in fact correct to state that heart weight and ventricular dimension are not influenced by formalin fixation (line 37). The data show quite a consistent change (3-7% based on the means) in dimensions between the fresh specimen and 24 hours. The study actually has two endpoints, the immediate response of the tissues over 24 hours and the subsequent difference between 24 and greater than 72 hours, and these go in both directions. The authors reference but do not attempt to interpret this difference. There IS an effect of formalin fixation but it occurs very rapidly after immersion in the solution.
This effect appears to be consistent and appears to reflect expansion in tissue dimensions and related reduction in chamber size. It does appear that dimensions did not change further after 24 hours. However, this is not the same as saying that there is no effect of fixation. Reference is also frequently made to statistical and clinical significance elsewhere in the paper. Neither of these is the same as biological significance or significance in contexts other than those of gross postmortem examination, while clinical significance must be considered contextually. For example, the variation observed between zero and 24 hours may have great significance to findings on histopathology or ultrastructural examination. Such findings should not be ignored. The potential ramifications of findings do not receive sufficient attention in the paper. Reference is made to the need for specialist involvement in the examination of the heart. It is doubtful any specialist would use weights derived without careful preparation of the organ, including removal of blood and adventitious tissue, or without careful consideration of clinical context.
Line Since drying appears to be a significant potential source of error, it would be helpful if more detail were provided on precisely how you dried your specimens and on how you would recommend drying be standardized. Equally, perhaps describe the appearance of an adequately dried specimen.
From the information you provide, the concentration of the formalin used would appear to be of significance. While you refer to this and report findings consistent with an impact of formalin immersion, little information is provided on the possible relationship between concentration and influences on structure and weight. I assume this is at least in part related to the osmotic potential of the fluid (tonicity) in the short term. A similar impact can apply when washing a sample with tap water. Bearing in mind the forensic context of the paper, more attention to the issue of the relative tonicity of fluids involved in tissue processing would be helpful.
You reference the use of tests for normality but it isn't clear in your presentation of data which variables were found to be normally distributed - please provide further detail.
Which ANOVA procedure was used, how was the issue of repeated measures addressed?
Please provide more information on how and when body weight was measured.
Since these were postmortem specimens and you indicate not all hearts were normal, at least in terms of the myocardium, the paper should indicate whether the response of the abnormal hearts was comparable to that of the normal hearts. This is especially important since you reference in your introduction a difference between normal myocardium and cardiomyopathy as being potentially significant.
The words didactic and didactical, while they should be immediately recognized and understood by most readers, might be replaced by "teaching" to optimize the clarity of the paper.
Clarification on what is meant by "time 0" is required - presumably this is tissue dimension after washing and drying and before immersion in formalin, i.e., measurement of the fresh specimen.
Probably of greater import in this paper than the overall postmortem procedure might be the process by which the heart was removed from the rest of the thoracic organs ("pluck"), if detail is to be provided. Recognizing that different tissues may show different responses to storage in formalin solution, one might argue, for example, that leaving pieces of a fatty pericardium in place might have a significant effect on weight change. It would be sufficient to say something like "the heart was removed from the thoracic cavity, the pericardium was removed and the great vessels were transected at the pericardial reflections”, or similar.
Janus et al., 2023, are referenced for details of the measurement technique employed, but that source does not provide sufficient information. The image presented in the referenced previous work clearly shows the choice of location for measurement is likely to be subjective. The technique employed here of marking the tissue location for repeated measurement would have helped immensely, but in the presence of changes in tissue dimension (see tables) and possible accompanying changes in tissue shape, this may have helped only partially. Though some measurements were repeated, a designed repeated measures procedure does not appear to have been employed. Can the authors be sufficiently confident that a significant though perhaps small difference was not present within the error involved in the measurement procedure to say that no progressive change was present or that such a change would have had no significance beyond gross examination?
Line 29 - "presently". Line 220. The expression "up to date" is used frequently. I think here, and similarly elsewhere, you may mean "at the present time" or "no such comparison has been published".
Line 97. "The weight of the heart was determined to the nearest gram."
Line 97, 108. "Weighed" rather than "weighted".
Line 112. "…measured using a manual caliper to the nearest 0.1 mm…". The ability to measure to the nearest 0.1 mm using manual calipers is questionable unless the calipers had a Vernier scale and the pressure applied during measurement was standardized. Was the person performing the measurements blinded to the previous measurement? Perhaps a morphometric technique that involves measuring the surface area of the section, possibly using an imprint of the section or a scanned image would have been preferable? As it is, the relationship between error and true change is not apparent. The authors might address the question in limitations.

Line 114. This should more properly read "… to ensure consistency in choosing the point of measurement" - this is not the same as repeatability, which the authors address below.
Line 118. Does this mean two additional researchers not otherwise involved in the project or does it mean two members of the project team? This technique does not minimize human error, though it might provide some opportunity to determine the contribution of individuals performing the measurements to overall variation.
Line 136. Can you please provide more information on "an incorrect angle". If you selected hearts for measurement on the basis of whether or not the tissues had been correctly prepared, this should perhaps be the "n" that you use for the paper, not the total number of hearts initially available. Alternatively, the fact that available tissues were limited by processing errors during teaching exercises might be explicitly stated early in the manuscript.
Figures. Why are the measurements staggered against the X axis in figure 6? The text implies not all measurements were made at the same time and that the authors considered this differences to be important. Was any analysis performed?
The figure images are far too small to read. If their inclusion is important, then perhaps they should be larger so they can be read.
Figure 2. In the review copy I was unable to enlarge the images, which are quite small, and some of the detail cannot be read. Enlargement would be helpful.
Comments on the Quality of English LanguageVery good use of English, occasional error in application of vernacular phrases
Author Response
Dear Reviewer,
Thank you very much for taking the time to review this manuscript. We believe that with the revisions made after all Reviewers' suggestions, the manuscript is improved. Please find the detailed responses below and the corresponding revisions/corrections highlighted red in the re-submitted files.
This paper describes changes in heart weight and cardiac wall and chamber dimensions in response to immersion in a 7% formalin solution and follows those changes for 72-96 hours. The objective of the investigation is to determine whether formalin preservation of cardiac tissues would materially impact findings for samples submitted for further investigation to referral centres. The paper is well-written, succinct and generally easy to follow. There are a number of items that are missing or inadequately described, however, while the focus of the paper and the authors' interpretations and conclusions seem to this reviewer to be more narrow than can be justified by the study's findings. Some information and the full implications of findings are missing.
It is not in fact correct to state that heart weight and ventricular dimension are not influenced by formalin fixation (line 37). The data show quite a consistent change (3-7% based on the means) in dimensions between the fresh specimen and 24 hours. The study actually has two endpoints, the immediate response of the tissues over 24 hours and the subsequent difference between 24 and greater than 72 hours, and these go in both directions. The authors reference but do not attempt to interpret this difference. There IS an effect of formalin fixation but it occurs very rapidly after immersion in the solution.
Thank you for that comment. We agree that there is a difference in cardiac dimensions prior and post fixation but it does not reach either statistical or clinical relevance. The aim of the study was to evaluate if feline and small-to-medium sized dogs’ hearts can be referenced for cardiopathological examination and if the fixation and shipment affects the clinical or pathological outcome in terms of gross evaluation. We have modified the aim of the study, the hypothesis, and expanded the discussion to make it clearer. Nonetheless, in the introduction we only added “significantly” in line 37, as here we refer to statistical analysis and the Introduction section does not allow more in-depth analysis.
This effect appears to be consistent and appears to reflect expansion in tissue dimensions and related reduction in chamber size. It does appear that dimensions did not change further after 24 hours. However, this is not the same as saying that there is no effect of fixation. Reference is also frequently made to statistical and clinical significance elsewhere in the paper. Neither of these is the same as biological significance or significance in contexts other than those of gross postmortem examination, while clinical significance must be considered contextually. For example, the variation observed between zero and 24 hours may have great significance to findings on histopathology or ultrastructural examination. Such findings should not be ignored. The potential ramifications of findings do not receive sufficient attention in the paper. Reference is made to the need for specialist involvement in the examination of the heart. It is doubtful any specialist would use weights derived without careful preparation of the organ, including removal of blood and adventitious tissue, or without careful consideration of clinical context.
Thank you for that comment. We agree that formalin fixation (even short term <24h) has an impact on tissue structure and further histopathological or ultrastructural examination. Nonetheless, exploring that impact was not the aim of the study. The aim of the study was to explore the impact of formalin fixation on cardiac weight and dimensions in the context of pathological and forensic examination performed within the first days of fixation. In that context we provided evidence that the post fixation examination and measurements are reliable. We added a short information in the discussion section on the possible impact of flushing, fixation and processing protocol on further analysis, nonetheless, in our opinion, discussing the topic is out of the scope of our paper.
Line Since drying appears to be a significant potential source of error, it would be helpful if more detail were provided on precisely how you dried your specimens and on how you would recommend drying be standardized. Equally, perhaps describe the appearance of an adequately dried specimen.
We added the details of tissue drying method with appropriate reference.
From the information you provide, the concentration of the formalin used would appear to be of significance. While you refer to this and report findings consistent with an impact of formalin immersion, little information is provided on the possible relationship between concentration and influences on structure and weight. I assume this is at least in part related to the osmotic potential of the fluid (tonicity) in the short term. A similar impact can apply when washing a sample with tap water. Bearing in mind the forensic context of the paper, more attention to the issue of the relative tonicity of fluids involved in tissue processing would be helpful.
The impact of various formalin concentration on tissue was already provided in the discussion section. As we did not evaluate the impact of various concentrations, we feel that deeper discussion (also regarding fluid chemical and physical aspects) would be excess.
You reference the use of tests for normality but it isn't clear in your presentation of data which variables were found to be normally distributed - please provide further detail.
All data showed normal distribution. We added information in the Materials and methods section.
Which ANOVA procedure was used, how was the issue of repeated measures addressed?
We used One-way repeated measures ANOVA - the information was added in the Materials and methods section.
Please provide more information on how and when body weight was measured.
The information was added in the Materials and Methods section.
Since these were postmortem specimens and you indicate not all hearts were normal, at least in terms of the myocardium, the paper should indicate whether the response of the abnormal hearts was comparable to that of the normal hearts. This is especially important since you reference in your introduction a difference between normal myocardium and cardiomyopathy as being potentially significant.
The tissues were not subjected to histopathological examination. The cardiac hypertrophy diagnosis was based on elevated cardiac weight and cardiac weight-to-body weight ratio. The results of the abnormal hearts were comparable to the normal ones.
The words didactic and didactical, while they should be immediately recognized and understood by most readers, might be replaced by "teaching" to optimize the clarity of the paper.
We changed the terms “didactic” and “didactical” to “teaching”, as suggested.
Clarification on what is meant by "time 0" is required - presumably this is tissue dimension after washing and drying and before immersion in formalin, i.e., measurement of the fresh specimen.
The clarification was added in the Materials and methods section.
Probably of greater import in this paper than the overall postmortem procedure might be the process by which the heart was removed from the rest of the thoracic organs ("pluck"), if detail is to be provided. Recognizing that different tissues may show different responses to storage in formalin solution, one might argue, for example, that leaving pieces of a fatty pericardium in place might have a significant effect on weight change. It would be sufficient to say something like "the heart was removed from the thoracic cavity, the pericardium was removed and the great vessels were transected at the pericardial reflections”, or similar.
Thank you for that comment but the information was already provided in the Materials and methods section (lines 100-102).
Janus et al., 2023, are referenced for details of the measurement technique employed, but that source does not provide sufficient information. The image presented in the referenced previous work clearly shows the choice of location for measurement is likely to be subjective. The technique employed here of marking the tissue location for repeated measurement would have helped immensely, but in the presence of changes in tissue dimension (see tables) and possible accompanying changes in tissue shape, this may have helped only partially. Though some measurements were repeated, a designed repeated measures procedure does not appear to have been employed. Can the authors be sufficiently confident that a significant though perhaps small difference was not present within the error involved in the measurement procedure to say that no progressive change was present or that such a change would have had no significance beyond gross examination?
Thank you for that comment. Nonetheless, the measures were not subjective; the measuring points were set as depicted in Fig 2D, perpendicularly to the ventricular walls, in the middle between the papillary muscles and carefully marked with tissue marker. The measurements were conducted by two independent researchers and repeated, in cases of differences between observers, the measurements were repeated; the mean value of four measurements (two repetitions made by two researchers) was calculated. In our opinion, it minimizes the possible error in sufficient manner.
Line 29 - "presently". Line 220. The expression "up to date" is used frequently. I think here, and similarly elsewhere, you may mean "at the present time" or "no such comparison has been published".
We made corrections as suggested.
Line 97. "The weight of the heart was determined to the nearest gram."
We made corrections as suggested.
Line 97, 108. "Weighed" rather than "weighted".
We made corrections as suggested.
Line 112. "…measured using a manual calliper to the nearest 0.1 mm…". The ability to measure to the nearest 0.1 mm using manual callipers is questionable unless the callipers had a Vernier scale and the pressure applied during measurement was standardized. Was the person performing the measurements blinded to the previous measurement? Perhaps a morphometric technique that involves measuring the surface area of the section, possibly using an imprint of the section or a scanned image would have been preferable? As it is, the relationship between error and true change is not apparent. The authors might address the question in limitations.
The measurements were taken using Vernier calliper. Both authors are experienced in using the calliper and performed the measurements without using additional pressure on the tissue and blinded to previous measurement. The information was added to the manuscript and as a limitation of the study.
Line 114. This should more properly read "… to ensure consistency in choosing the point of measurement" - this is not the same as repeatability, which the authors address below.
We made corrections as suggested.
Line 118. Does this mean two additional researchers not otherwise involved in the project or does it mean two members of the project team? This technique does not minimize human error, though it might provide some opportunity to determine the contribution of individuals performing the measurements to overall variation.
The measurements were performed by both authors - we clarified the information in the manuscript.
Line 136. Can you please provide more information on "an incorrect angle". If you selected hearts for measurement on the basis of whether or not the tissues had been correctly prepared, this should perhaps be the "n" that you use for the paper, not the total number of hearts initially available. Alternatively, the fact that available tissues were limited by processing errors during teaching exercises might be explicitly stated early in the manuscript.
We explained the term “incorrect angle” in the Results section and added information about tissue exclusion in the Materials and methods section. We chose “n” for the study as the exclusion groups did not totally overlap and therefore 90 hearts served for the comparison of the organ’s weight prior and after blood flushing and tissue drying and 74 hearts served for the comparison of ventricular dimensions. All hearts (134) served for the comparison of organ weight in study time points. The total number of hearts used in the study was 134 hearts.
Figures. Why are the measurements staggered against the X axis in figure 6? The text implies not all measurements were made at the same time and that the authors considered this differences to be important. Was any analysis performed?
Unfortunately, we do not fully understand this comment. The measurements were performed on each heart after flushing and drying (fresh sample - time point 0), and then repeated after 24, 48, 72 and >72h. It is presented on X axis. The statistical software visualises the results for each dimension with a little slide to prevent overlapping of the whiskers. We cannot address the comment that we consider the difference in time of measurements as important as we cannot find such statement in the text.
The figure images are far too small to read. If their inclusion is important, then perhaps they should be larger so they can be read.
The size and resolution of images is probably reduced during pdf preparation for the review process. We tried to improve the images and still believe that, if accepted, the manuscript will include the original figures with good resolution and size
Figure 2. In the review copy I was unable to enlarge the images, which are quite small, and some of the detail cannot be read. Enlargement would be helpful.
As answered above.
Reviewer 3 Report
Comments and Suggestions for Authors
The impact of short-term formalin-fixation on the weight and 2 ventricular dimensions in the hearts of cats and small-to-me-3 dium-sized dogs 4
Izabela Janus-ZióÅ‚kowska 1,* and Joanna Bubak 2
The reviewer thanks the authors for this nice and valuable study.
This is important information worth sharing with the veterinary community.
Please see the comments below to add more clarity to the study.
Specific Comments:
Line 11: Please change to: Post mortem studies help …
Line 15 : Hearts not heart
Line 16: show not shown
Line 22: to obtain the opinion of a specialist, …
Line 24: size are important data (strike “the”)
Line 45 : or suspicion of hypertrophic cardiac myopathy
Line 48 : change to: In veterinary medicine no similar recommendation exists even though there are species breed and size differences that warrant poste mortem examination by a specialist. – Please strike : old line 48 In veterinary medicine to Line 51 referral opinions.
Line 55 hearts
Line 80 you should state a hypothesis: did you think formalin fixation does or does not affect the weight of the heart? – Please add your original hypothesis.
Material and Methods,
Please explain who did the necropsy, who measured the heard and what was the standard of dryness you used. Also how long were the patient’s diseased before the necropsy was done. Please provide more detail here.
Line 200: add value behind the word borderline
Line 206 – Please reword to exclude the tumour reference. You are looking at heart size and heart disease not tumours.The paragraph Line 211 to Line 221 is not helpful to the study. Referencing tumour tissue is not ideal.
Line 222: this and the next paragraph help to understand the study.
It would be noce to get more of an understanding why only small to medium size dogs were chosen vs large dogs. Please expand on the discussion. Line 248, please change the line to …these assumptions on large- or giant breed dogs.
Author Response
Dear Reviewer,
Thank you very much for taking the time to review this manuscript. We believe that with the revisions made after all Reviewers' suggestions, the manuscript is improved. Please find the detailed responses below and the corresponding revisions/corrections highlighted red in the re-submitted files.
The reviewer thanks the authors for this nice and valuable study.
This is important information worth sharing with the veterinary community.
Please see the comments below to add more clarity to the study.
Specific Comments:
Line 11: Please change to: Post mortem studies help …
We made corrections as suggested.
Line 15 : Hearts not heart
We made corrections as suggested.
Line 16: show not shown
We made corrections as suggested.
Line 22: to obtain the opinion of a specialist, …
We made corrections as suggested.
Line 24: size are important data (strike “the”)
We made corrections as suggested.
Line 45 : or suspicion of hypertrophic cardiac myopathy
We rephrased the sentence; it is important to examine the heart in all cases of cardiac hypertrophy, not only hypertrophic cardiomyopathy.
Line 48 : change to: In veterinary medicine no similar recommendation exists even though there are species breed and size differences that warrant poste mortem examination by a specialist. – Please strike : old line 48 In veterinary medicine to
We made corrections as suggested.
Line 51 referral opinions.
We made corrections as suggested.
Line 55 hearts
We made corrections as suggested.
Line 80 you should state a hypothesis: did you think formalin fixation does or does not affect the weight of the heart? – Please add your original hypothesis.
Thank you for that comment. We added the hypothesis.
Material and Methods,
Please explain who did the necropsy, who measured the heard and what was the standard of dryness you used. Also how long were the patient’s diseased before the necropsy was done. Please provide more detail here.
Thank you for that comment. We added more details in the Materials and methods section.
Line 200: add value behind the word borderline
We made corrections as suggested.
Line 206 – Please reword to exclude the tumour reference. You are looking at heart size and heart disease not tumours.The paragraph Line 211 to Line 221 is not helpful to the study. Referencing tumour tissue is not ideal.
We believed that it added value to the discussion, but removed the tumour-related information, as suggested.
Line 222: this and the next paragraph help to understand the study.
Thank you for that comment.
It would be noce to get more of an understanding why only small to medium size dogs were chosen vs large dogs. Please expand on the discussion.
Thank you for that comment. We added the information in the limitation part of the discussion.
Line 248, please change the line to …these assumptions on large- or giant breed dogs.
We made corrections as suggested.
Round 2
Reviewer 2 Report
Comments and Suggestions for Authors
Line 77, revised manuscript. This paragraph, although revised, is still not appropriate. It contradicts concerns expressed by this reviewer and the authors' response thereto. A small but critical detail. I am particularly concerned about the insertion of the term "clinically significant", because this study did not, and indeed could not assess clinical significance, which is opinion-based. I suggest the last paragraph be replaced with the following which I present for the authors to consider…
"The aim of this study was to evaluate the effect of short term (72 hour) formalin fixation on cardiac weight and ventricular dimensions in cats and small to medium-sized dogs to determine whether there is a statistically significant impact of processing.”
Line 90. With a precision of 0.1 kg, or alternatively to the nearest 0.1 kg.
Author Response
Dear Reviewer,
Thank you for additional comments. Please find the detailed responses below and the corresponding corrections highlighted red in the re-submitted file.
Line 77, revised manuscript. This paragraph, although revised, is still not appropriate. It contradicts concerns expressed by this reviewer and the authors' response thereto. A small but critical detail. I am particularly concerned about the insertion of the term "clinically significant", because this study did not, and indeed could not assess clinical significance, which is opinion-based. I suggest the last paragraph be replaced with the following which I present for the authors to consider…
"The aim of this study was to evaluate the effect of short term (72 hour) formalin fixation on cardiac weight and ventricular dimensions in cats and small to medium-sized dogs to determine whether there is a statistically significant impact of processing.”
Thank you for that comment. We changed the paragraph as suggested. Also, we left the second part of the paragraph stating the study hypothesis, as it was suggested by another Reviewer.
Line 90. With a precision of 0.1 kg, or alternatively to the nearest 0.1 kg.
We made correction as suggested.